# Efficacy of Combination Therapies for the Treatment of Multi-Drug Resistant Gram-Negative Bacterial Infections Based on Meta-Analyses

**DOI:** 10.3390/antibiotics11040524

**Published:** 2022-04-14

**Authors:** Takumi Umemura, Hideo Kato, Mao Hagihara, Jun Hirai, Yuka Yamagishi, Hiroshige Mikamo

**Affiliations:** 1Department of Clinical Infectious Diseases, Aichi Medical University, Nagakute 480-1195, Japan; umemuratakumi@gmail.com (T.U.); katou.hideo.233@mail.aichi-med-u.ac.jp (H.K.); hagimao@aichi-med-u.ac.jp (M.H.); hiraichimed@gmail.com (J.H.); y.yamagishi@mac.com (Y.Y.); 2Department of Molecular Epidemiology and Biomedical Sciences, Aichi Medical University, Nagakute 480-1195, Japan

**Keywords:** antibiotics, combination therapy, multi-drug resistant infection, meta-analysis

## Abstract

There is increasing evidence regarding the optimal therapeutic strategies for multidrug-resistant (MDR) bacteria that cause common infections and are resistant to existing antibiotics. Combination therapies, such as β-lactam combined with β-lactamase inhibitors or combination antibiotics, is a therapeutic strategy to overcome MDR bacteria. In recent years, the therapeutic options have expanded as certain combination drugs have been approved in more countries. However, only a handful of guidelines support these options, and the recommendations are based on low-quality evidence. This review describes the significance and efficacy of combination therapy as a therapeutic strategy against Gram-negative MDR pathogens based on previously reported meta-analyses.

## 1. Introduction

Antibiotic resistance is one of the top ten global public health threats. The World Health Organization (WHO) has reported that there is mounting evidence that the spread of multidrug-resistant (MDR) bacteria that cause common infections and are resistant to treatment with existing antibiotics is increasing [1,2]. In recent years, only a few new antibiotics have been developed for the treatment of infections by MDR pathogens, and emergent resistance has been reported after clinical use [3]. Therefore, the Center for Disease Prevention and Control (CDC) identified MDR Gram-negative bacteria such as carbapenem-resistant Acinetobacter and carbapenem-resistant Enterobacterales (CRE) as urgent threats because the rapid spread of resistance and lack of appropriate treatment strategies are considered critical [4].

Combination therapy is a strategy for preventing infections caused by MDR Gram-negative pathogens [5,6,7,8]. New combinations are increasingly proposed as a therapeutic option with the increasing approval of novel drugs. The combinations include antibiotics plus drugs without antibiotic activity, or antibiotics plus other antibiotics. However, only a handful of guidelines support this option, and they are based on low-quality evidence [9], since there are few randomized controlled trials in the literature that provide high levels of evidence to support the clinical question.

Meta-analyses can contribute to the establishment of evidence-based strategies and resolve contradictory research outcomes. In fact, many researchers have compared combination therapy with monotherapy using the meta-analysis. However, the interpretation of combination therapy may be misleading since criteria such as infection type and pathogen are different among these studies. Therefore, we comprehensively summarize published meta-analyses and illustrate the significance and feature of combination therapy as a therapeutic option against MDR pathogens. Table 1 shows a list of studies that reported meta-analyses of antibiotic combinations [10,11,12,13,14,15,16,17,18,19,20,21,22,23,24].

## 2. Combinations of Antibiotics plus β-Lactamase Inhibitor

β-lactamase production is one of the main mechanisms of resistance against β-lactams. To date, many types of β-lactamases have been reported: Ambler class A, which includes TEM nova (TEM), sulfhydryl variable (SHV), cefotaxime (CTX), *Klebsiella pneumoniae* carbapenemases (KPC), and *Serratia marcescens* enzymes (SME); serine-based class B, which includes Verona integron-encoded metallo-β-lactamase (VIM), imipenemase metallo-β-lactamase (IMP), New Delhi metallo-β-lactamase (NDM), and Sao Paulo metallo-β-lactamase (SPM); zinc-based class C, which includes AmpC; and serine-based class D, which includes oxacillinase (OXA).

Recently, the Infectious Diseases Society of America (IDSA) published a guideline recommending carbapenems as the preferred treatment option for infections caused by resistant bacteria, without mentioning β-lactam and β-lactamase inhibitor combination therapies [26]. However, a recent study based on the MERINO trial reported that differences in mortality rates are less pronounced for β-lactams used in combination with β-lactamase inhibitors [25]. Moreover, several novel β-lactam and β-lactamase inhibitor combination therapies have been developed that target Gram-negative bacteria that produce β-lactamase. Therefore, β-lactam and β-lactamase inhibitor combination therapies may be a novel alternative option for the treatment of β-lactamase-producing pathogens.

### 2.1. Carbapenem versus β-Lactam and β-Lactamase Inhibitor Combinations

Extended-spectrum β-lactamases (ESBL) are class A β-lactamase enzymes that hydrolyze the β-lactam ring, conferring resistance to most β-lactam antibiotics, including expanded-spectrum cephalosporins, and are often resistant to other classes of antibiotics (e.g., fluoroquinolones, trimethoprim-sulfamethoxazole, aminoglycosides, and tetracyclines) [27,28,29,30]. Moreover, the overuse of carbapenems, which are generally recommended as first-line agents for infections caused by ESBL-producing pathogens [26], lead to an increase in selective pressures for carbapenem resistance [31]. Therefore, the interest in β-lactam and β-lactamase inhibitor combination therapies has increased due to the limited treatment options for infections caused by ESBL-producing pathogens.

Two meta-analyses have evaluated the efficacy of carbapenem versus β-lactam and β-lactamase inhibitor combination therapies for the treatment of infections caused by ESBL-producing pathogens (Table 2). Sfeir et al. conducted a meta-analysis to investigate the 30-day mortality of patients with blood stream infections caused by ESBL-producing pathogens [10]. A total of twenty-five cohort or case-control studies were included: eleven evaluated empiric treatment, eight evaluated definitive treatment, and six simultaneously evaluated empiric and definitive treatment. No statistically significant differences in mortality were found between β-lactam and β-lactamase inhibitor combination therapy and carbapenem administered as an empirical (odds ratio [OR] 1.13, 95% confidence interval [CI] 0.87–1.48) or definitive (OR 0.96, 95% CI 0.50–1.86) treatment. In subgroup analyses, there was no significant difference in mortality between patients treated with piperacillin-tazobactam (PTZ) and carbapenem when used as empirical (OR 1.26, 95% CI 0.96–1.66) or definitive (OR 0.97, 95% CI 0.59–1.6) treatment for ESBL-producing Enterobacterales bloodstream infections. Zhang et al. compared clinical outcomes, such as clinical success, microbiological success, and mortality for urinary tract infections (UTIs) caused by ESBL-producing Enterobacterales [11]. Three randomized controlled trials (RCTs) and seven cohort studies were included. There was no statistically significant difference between carbapenem and β-lactam and β-lactamase inhibitor combination therapy with regards to clinical success (risk rate [RR] 0.99, 95% CI 0.96–1.03) and mortality (RR 0.63, 95% CI 0.30–1.32). In contrast, a slightly higher rate of microbiological success was observed in patients treated with a β-lactam and β-lactamase inhibitor combination therapy (RR 1.06, 95% CI 1.01–1.11). However, this result was mainly attributed to treatment with ceftazidime-avibactam (CZA) based on a single RCT (RR 1.32, 95% CI 1.13–1.55).

### 2.2. Carbapenem versus Ceftazidime-Avibactam

CZA is a combination of the third-generation cephalosporin ceftazidime and the novel non-β-lactam β-lactamase inhibitor avibactam. CZA shows in vitro activity against Amber class A, class C, and some class D β-lactamase-producing bacteria, including Enterobacterales and *Pseudomonas aeruginosa*, whereas it has no activity against metallo-β-lactamase-producing organisms [32,33,34]. CZA has been approved by the United States Food and Drug Administration (FDA) and the European Medicines Agency (EMA) for the treatment of infections caused by Gram-negative bacteria for which limited therapeutic options exist [35,36].

There have been three meta-analyses on the efficacy and safety of carbapenem versus CZA for infections caused by Enterobacterales (Table 3). Sternbach et al. assessed the efficacy and safety of this treatment for complicated infections [12]. Eligible studies included RCTs for the treatment of complicated UTIs, complicated intra-abdominal infections, and nosocomial pneumonia among adult patients. Seven RCTs were included, of which less than 25% of cases were ESBL-producing Enterobacterales at the initiation of treatment. All-cause 30-day mortality was reported in six trials, which included one UTI, four intra-abdominal infection, and one pneumonia trial, that showed no significant difference in response between carbapenem and CZA (RR 1.10, 95% CI 0.70–1.72). Moreover, a microbiological response was reported in five trials, including three UTI, one intra-abdominal infection, and one pneumonia trial, that showed no significant difference between carbapenem and CZA (RR 1.04, 95% CI 0.93–1.17). In contrast, serious adverse events were reported in six trials including three UTI, two intra-abdominal infections, and one pneumonia trial, and a significantly higher rate of discontinuation was demonstrated for patients treated with CZA (RR 1.24, 95% CI 1.00–1.54). Che et al. evaluated the feasibility of treating Enterobacterales infections with CZA instead of carbapenem [13]. In this meta-analysis, eligible studies included RCTs that treated adult patients for Enterobacterales infections or mixed infections with Enterobacterales bacteria accounting for more than 90% of infections in the population, and three RCTs were included. The meta-analysis of the three trials showed that there were no significant differences between CZA and carbapenems in the rate of clinical success [risk difference (RD) 0.00, 95% CI −0.06–0.06] and microbiological success (RD 0.07, 95% CI −0.04–0.18). In contrast, serious adverse events were reported in two studies and tended to occur more frequently in patients treated with CZA than in those treated with carbapenems (RD 0.02, 95% CI −0.00–0.04). Isler et al. conducted a meta-analysis of adult patients who were suspected to have or were diagnosed with infection with ESBL- or AmpC-producing Enterobacterales in complicated UTI or complicated intra-abdominal infection and pneumonia [14]. Five RCTs were available to evaluate clinical and microbiological responses; four studies reported outcome data for ESBL-producing Enterobacterales; three studies reported outcome data for AmpC-producing Enterobacterales; and four studies reported outcome data for ceftazidime non-susceptible Enterobacterales. The clinical response for ESBL- or AmpC-producing Enterobacterales showed no significant differences between patients treated with carbapenem and CZA (clinical response for ESBL producers, RR 1.02, 95% CI 0.97–1.08; for AmpC producers, RR 0.91, 95% CI 0.76–1.10). CZA showed a better microbiologic response than carbapenem for ceftazidime non-susceptible Enterobacterales in patients with complicated UTI and pneumonia (RR 1.21, 95% CI 1.07–1.37).

### 2.3. CZA versus CZA Combination Therapy

CZA monotherapy led to the emergence of resistance-conferring mutations in blaKPC-3 in *K. pneumoniae* [37]. Moreover, a recent study reported that *P. aeruginosa* that produce metallo-β-lactamases (VIM, IMP, and NDM) have a high resistance rate against CZA [38]. Therefore, CZA monotherapy may not be sufficient to treat all MDR bacteria. However, combination therapy with CZA is expected to be a therapeutic strategy in patients infected with MDR bacteria, since in vitro studies have shown positive effects with CZA plus other antibiotics [39,40].

Three meta-analyses compared CZA monotherapy with CZA combination therapy in patients with carbapenem-resistant Gram-negative pathogens (Table 4). Onorato et al. conducted a meta-analysis of infections caused by CRE and carbapenem-resistant *P.*
*aeruginosa* [15]. Eleven retrospective studies were included, and three were case series. Seven studies included only patients with CRE, one included only patients with carbapenem-resistant *P. aeruginosa*, and three reported infections caused by both pathogens. The meta-analysis included various types of infections, including pneumonia, bacteremia, intra-abdominal infection, UTI, skin and soft-tissue infection, bone and joint infection, and multiple infections. Six studies included patients with a KPC-producing strain, five studies included patients with an OXA-48-producing strain, and the strain of resistance was unknown in some patients. The mortality rate was similar in patients treated with monotherapy and combination therapy (RR 1.18, 95% CI 0.88–1.58). Similarly, the mortality rate was not significantly different between the two groups in five studies evaluating in-hospital mortality (RR 1.37, 95% CI 0.80–2.34), four studies evaluating 30-day mortality (RR 1.07, 95% CI 0.75–1.53), and two studies evaluating 90-day mortality (RR 1.42, 95% CI 0.44–4.60). Regarding microbiological outcome, no difference was observed between the two groups in seven studies (RR 1.04, 95% CI 0.85–1.28). In three studies microbiological outcome was defined as the presence of at least one negative culture during therapy (RR 1.02, 95% CI 0.67–1.56), in two studies microbiological outcome was defined as negative cultures after more than 7 days of treatment (RR 0.92, 95% CI 0.87–1.97), and in six studies patients infected with carbapenem-resistant *P. aeruginosa* (RR 1.05, 95% CI 0.84–1.31) were excluded. Fiore et al. conducted a network meta-analysis of patients with CRE infections [16]. Six retrospective studies were included, and mortality was reported at different time points. The network meta-analysis showed no significant difference in the mortality rate between patients who received CZA monotherapy and those who received CAZ combination therapy (OR 0.96, 95% CI: 0.65–1.41). Li et al. conducted a meta-analysis to compare mortality rate, microbiological response, clinical success, and development of resistance [17]. Seventeen retrospective observational studies were included: eleven cohort studies, one case-cohort study, two case series, and three cross-sectional studies. The pathogen reported by most of these studies was CRE, with carbapenem-resistant *K. pneumoniae* as the most frequently reported pathogen, while one study included both carbapenem-resistant *K. pneumoniae* and *P. aeruginosa*, and one study included CRE, carbapenem-resistant *P. aeruginosa*, and *A. baumannii*. Regarding infection types, most studies reported multiple infections, two studies reported only bacteremia, and one study reported pneumonia. Antibiotics that were used in combination with CZA therapy included aminoglycosides, polymyxin, tigecycline, colistin, amikacin, imipenem, gentamicin, ciprofloxacin, meropenem, fosfomycin, carbapenems, fluoroquinolones, minocycline, and sulfamethoxazole-trimethoprim. There was no statistically significant difference in mortality rate at any time point (overall, 14, 30, and 90 days, in-hospital) between CZA therapy alone and CZA-based combination therapy (overall, OR 1.03, 95% CI 0.79–1.34; 14-day, OR 0.90, 95% CI 0.32–2.50; 30-day, OR 0.96, 95% CI 0.69–1.33; 90-day, OR 1.74, 95% CI 0.79–3.82; in-hospital, OR 1.01, 95% CI 0.55–1.86). No statistically significant difference was found between groups in the microbiological response in five studies (OR 0.99, 95% CI 0.54–1.81), in the clinical success in ten studies (OR 0.95, 95% CI 0.64–1.39), or in the post-treatment resistance to CZA in six studies (OR 0.65, 95% CI 0.34–1.26). However, the combination therapy was more strongly associated with lower resistance to CZA in the three pooled studies (OR 0.18, 95% CI 0.04–0.78).

### 2.4. Ceftlozane-Tazobactam (C/T) versus C/T Combination Therapy

Tazobactam targets the active site of serine-based β-lactamases, which are mainly class A β-lactamases [18]. Ceftlozane, an oxyamino-aminothiazolyl cephalosporin, is structurally similar to ceftazidime and is active against *P. aeruginosa* [18]. A combination with ceftlozane and tazobactam has been approved for the treatment of complicated UTI, complicated intra-abdominal infections, and hospital-acquired and ventilator-associated pneumonia (HAP/VAP) [41,42,43]. In vitro studies evaluating combination regimens containing C/T plus other antibiotics showed a decline in the bacterial burden of MDR pathogens [44,45,46,47]. In contrast, clinical studies have shown discrepancies with the results of preclinical studies.

One meta-analysis reported a comparison between C/T alone and C/T in association with other antibiotics for the treatment of adult patients with microbiologically confirmed bacterial infections in any setting [48]. The study included seven retrospective cohort studies (two multicenter and five single-center) and one single-center case-control study. Seven of the eight studies evaluated infections caused by *P. aeruginosa* and the other evaluated infections caused by ESBL-producing Enterobacterales. Patients developed sepsis in two of the eight studies, lower respiratory tract infection in one study, and osteomyelitis in another study. Four studies that evaluated all-cause mortality enrolled 148 patients (C/T, 87 patients; C/T combination therapy, 61 patients), and the mortality rate was significantly decreased with C/T combination therapy compared to that with C/T monotherapy (OR 0.31, 95% CI 0.10–0.97, *p* = 0.045). Seven studies that enrolled 391 patients evaluated clinical improvement outcomes (C/T, 261 patients; C/T combination therapy, 130 patients), and the clinical outcome did not improve using C/T combination therapy (OR 0.97, 95% CI 0.54–1.74, *p* = 0.909). Two studies that enrolled 33 patients evaluated microbiological cure outcomes (C/T, 13 patients; C/T combination therapy, 20 patients), and there was no significant difference in microbiological cure between C/T combination therapy and C/T monotherapy (OR 0.83, 95% CI 0.12–5.70, *p* value was not reported). The discrepancy between the overall estimate of the effect between the mortality and clinical outcome (clinical cure and microbiologic cure) is difficult to interpret clinically, but heterogeneities in sample size and patient backgrounds may explain the difference.

## 3. Antibiotics Combinations

The IDSA guidelines do not recommend routine antibiotic combination therapy for infections caused by carbapenem-resistant Enterobacterales and *P. aeruginosa* [10] since the combination therapy increases the incidence of nephrotoxicity [19,20,49,50]. However, few reports have shown an increase in other adverse events caused by the combination therapy [6]. Currently, new β-lactam and β-lactamase inhibitor combination therapies, such as CZA, C/T, imipenem-cilastatin-relebactam, and meropenem-vaborbactam, have been developed to treat infections caused by MDR pathogens. Tazobactam targets the active site of serine-based β-lactamases, mainly class A β-lactamases such as SHV [33], and REL and VAB inhibit class A and C β-lactamases, but not class B and D [51,52]. Avibactam inhibits class A, C, and D serine-based β-lactamase inhibitors [33]. Notably, these β-lactam and β-lactamase inhibitor combinations are unable to treat all carbapenemase-producing pathogens. Therefore, attention has been focused on antibiotic combinations as treatment options for these pathogens. Here, we summarize antibiotic combinations for the treatment of infections caused by MDR Gram-negative bacteria based on previous meta-analyses.

### 3.1. β-Lactam versus β-Lactam plus Aminoglycoside

Antibiotic synergy has traditionally been demonstrated with β-lactam–aminoglycoside combinations for the treatment of infections with Gram-negative pathogens. The combination of β-lactams and aminoglycosides provides different mechanisms by which bacteria are eliminated [6]. However, clinical studies have shown results that are contrary to those of in vitro and in vivo studies.

Two meta-analyses have evaluated the efficacy of β-lactam versus β-lactam plus aminoglycoside (Table 5). Paul et al. performed a meta-analysis of RCTs on the treatment of patients with fever and neutropenia [19]. Forty-seven RCTs were included in this meta-analysis. Ceftazidime, PTZ, imipenem, and cefoperazone were administered as β-lactam regimens, while amikacin, gentamicin, and tobramycin were administered as aminoglycoside regimens. Although combination therapy did not improve all-cause mortality compared to monotherapy (RR 0.85, 95% CI 0.72–1.02), there was a significantly higher treatment success rate using combination therapy for the treatment of severe neutropenia (<100/mm3; RR 1.49, 95% CI 1.13–1.97) in both adults >16 years old (RR 1.21, 95% CI 1.07–1.37) and children (RR 2.74, 95% CI 1.08–6.98). In addition, Paul et al. performed a meta-analysis of RCTs for severe infections in patients without neutropenia [20]. Sixty-four RCTs were included in this meta-analysis. Infection types included severe sepsis, pneumonia, Gram-negative infections, abdominal infections, UTIs, and Gram-positive infections. Twelve RCTs reported all-cause mortality, and there was no difference between monotherapy and combination therapy (RR 1.02, 95% CI 0.76–1.38). Clinical and bacteriological failures were evaluated in data from twenty and fourteen RCTs, respectively, and combination therapy showed a tendency to improve clinical failure compared with same β-lactam monotherapy (RR 1.09, 95% CI 0.94–1.27), while there was no significant difference in bacteriological failure between the two groups (RR 1.08, 95% CI 0.71–1.64). Furthermore, nephrotoxicity was significantly more common with combination therapy (RR 0.36, 95% CI 0.28–0.47).

### 3.2. Carbapenem plus Carbapenem (Double Carbapenems) versus Other Antibiotic Regimens

Bulik et al. reported the efficacy of double carbapenem therapy (DCT) against carbapenemase-producing *K. pneumoniae* in a mouse thigh infection model [53]. Moreover, it was reported that DCT had a synergistic effect in isolates with a high minimum inhibitory concentration for meropenem (up to 128 mg/L) [54]. In 2013, DCT successfully cured three patients with KPC-producing *K. pneumoniae* [55]. Over the past few years, DCT has emerged as a promising treatment strategy for carbapenem-resistant *K. pneumoniae* [56].

One meta-analysis evaluated the efficacy and safety of DCT and other antibiotic regimens in patients with infections caused by MDR Gram-negative pathogens [57]. The study included three cohort or case-control studies comprising 235 patients with CRE infection, and the infection types mainly included pneumonia, bloodstream infections, and UTIs. The DCT regimens were combinations of ertapenem (1–2 g daily) and meropenem (2 g every 8 h daily) or doripenem (2 g every 8 h daily), and other antibiotic regimens were colistin, tigecycline, aminoglycoside monotherapies, or combination regimens. There were no obvious advantages of DCT in clinical response (OR 1.74, 95% CI 0.99–3.06, *p* = 0.05) or microbiological response (OR 1.90, 95% CI 0.95–3.80, *p* = 0.07), but the mortality rate in the DCT group was significantly lower than that in the control group (OR 0.44, 95% CI 0.24–0.82, *p* = 0.009). No adverse events resulted in treatment interruption. A critical limitation of this study was the low grade of evidence, because the analysis was based on only three retrospective cohort or case-control studies. However, current data suggest that DCT may be an effective and safe strategy for treating carbapenem-resistant pathogens.

### 3.3. Polymyxin versus Polymyxin Combination Therapy

The alarming increase in MDR Gram-negative bacteria has resulted in the resurgence of polymyxin use, although its use has been limited for the past decade [58]. Polymyxin is at times the only therapeutic option because of its variable susceptibility to other antibiotics [59]. However, the use of polymyxin has several disadvantages, including poor efficacy compared to β-lactam [59], nephrotoxicity induced by high doses [60], and the emergence of resistance during therapy [61]. Therefore, in clinical settings, polymyxin is used in combination with other antibiotics to improve clinical outcomes.

Two meta-analyses have evaluated the efficacy of polymyxin versus polymyxin combination therapy (Table 6). Zusman et al. performed a meta-analysis examining the effectiveness of polymyxin monotherapy versus polymyxin-based combination therapy by antibiotic type and bacterial species [21]. Eligible studies included retrospective studies, prospective studies, or RCTs in adult patients infected with polymyxin-susceptible, carbapenem-resistant, or carbapenemase-producing Gram-negative bacteria, and twenty-two studies were included: nineteen retrospective observational studies and three RCTs. Overall, nine studies assessed tigecycline, seven studies assessed carbapenems, three studies assessed rifampicin, three studies assessed aminoglycosides, three studies assessed sulbactam, two studies assessed vancomycin, one study assessed PTZ, and one study assessed intravenous fosfomycin. Mortality rates were significantly higher with polymyxin monotherapy compared to those with carbapenem combination therapy in seven studies (OR 1.58, 95% CI 1.03–2.42), and those with tigecycline, aminoglycoside, or fosfomycin in eleven studies (OR 1.57, 95% CI 1.06–2.32). In particular, seven studies on the treatment of *K. pneumoniae* bacteremia found that combination therapy with tigecycline or aminoglycoside resulted in significantly lower mortality rates than those with polymyxin monotherapy (OR 2.09, 95% CI 1.21–3.60). When monotherapy was compared to any combination therapy for the type of infection, any combination therapy was associated with improved mortality rates in the treatment of bacteremia (OR 2.23, 95% CI 1.51–3.30), while there was no significant difference in mortality rates between monotherapy and combination therapy for the treatment of ventilator-associated or hospital-acquired pneumonia in five studies (OR 0.69, 95% CI 0.39–1.24).

Samal et al. [22] performed a meta-analysis with similar eligibility criteria as the previously mentioned study [21]. This study included seventeen prospective studies (6 RCTs) and twenty-two retrospective studies. A meta-analysis of all-cause mortality in all 39 studies yielded an OR of 0.81 with a 95% CI of 0.65–1.01. Nine studies that used only carbapenems yielded an OR of 0.64 with a 95% CI of 0.40–1.03. Moreover, a separate meta-analysis of only RCTs yielded an OR of 0.82 with a 95% CI of 0.58–1.16.

### 3.4. Colistin versus Colistin Combination Therapy

Colistin and polymyxin B differ by a single amino acid in the peptide ring, with phenylalanine in polymyxin B and leucine in colistin [62]. Colistin is a mixture of colistin A and colistin B, with colistin B accounting for the majority of the total dose [63], and is administered as the inactive pro-drug colistimethate sodium [64]. The different degrees of protein binding between colistin A and B cause inter- and intra-individual variability in the plasma concentrations of colistin [65,66]. Several approaches have been proposed to overcome these problems. Among the implemented strategies, combination regimens have been identified as the most promising [67]. In vitro studies have demonstrated synergistic and additive effects when combination regimens including colistin were evaluated [68,69,70,71]. In particular, the combination of colistin and tigecycline has been found to be effective against strains that form biofilms, although this may be dependent on the concentration of each drug [72].

Recently, 3 meta-analyses have evaluated the efficacy of colistin versus colistin combination therapy (Table 7). Cheng et al. conducted a meta-analysis of five RCTs on the treatment of carbapenem-resistant Gram-negative bacterial infections [23]. All *A. baumannii* isolates were carbapenem-resistant. The colistin-based combination regimens included rifampicin (two studies), fosfomycin (one study), meropenem (one study), and ampicillin-sulbactam (one study). Compared to colistin combination therapy, colistin monotherapy had no association with higher mortality in five studies (RR 1.03, 95% CI 0.89–1.20), higher infection-related mortality in four studies (RR 1.23, 95% CI 0.91–1.67), and lower microbiologic response in five studies (RR 0.86, 95% CI 0.72–1.04). In addition, compared to colistin combination therapy, colistin monotherapy was not associated with lower nephrotoxicity in three studies (RR 0.98, 95% CI 0.84–1.21). Vardakas et al. conducted a meta-analysis of all study types, except case reports and case series, on MDR or extensively drug-resistant (XDR) Gram-negative infections [9]. Thirty-two studies were included: twenty-two retrospective studies, six prospective studies, one study with both prospective and retrospective aspects, and three RCTs. The studies focused mainly on infections caused by *A. baumannii* and *K. pneumoniae*; infections caused by *P. aeruginosa* and other Enterobacterales were also included. Carbapenem, tigecycline, gentamicin, rifampin, sulbactam, fosfomycin, non-carbapenem β-lactams, and ciprofloxacin were the main antibiotics used in the combination regimens in various proportions. The rates of resistance to individual antibiotics varied. Compared to colistin monotherapy, colistin-based combination therapy was not associated with lower mortality (RR 0.91, 95% CI 0.81–1.02). In sub-analyses of high-dose treatments (>6 million international units; RR 0.80, 95% CI 0.69–0.93), combination therapy was found to be significantly more effective in patients with bacteremia (RR 0.75, 95% CI 0.57–0.98), and in patients with *A. baumannii* infections (RR 0.88, 95% CI 0.78–1.00). Liu et al. performed a network meta-analysis of the treatment of MDR *A. baumannii* infections [24]. Eighteen studies were included: seven RCTs and eleven retrospective studies. Eleven studies focused on pneumonia, whereas the other studies included patients with mixed infections. Rifampicin, fosfomycin, sulbactam, and carbapenem were used in combination regimens with colistin. There was no significant difference in clinical improvement and cure between monotherapy and any combination therapy (clinical improvement: rifampicin, RR 1.28, 95% CI 0.67–2.45; fosfomycin, RR 1.08, 95% CI 0.76–1.53; sulbactam, RR 1.02, 5% CI 0.86–1.22; clinical cure: carbapenem, RR 1.34, 95% CI 0.92–1.95; sulbactam, RR 1.28, 95% CI 0.90–1.83). The combination of rifampicin and fosfomycin was associated with a significantly higher rate of microbiological eradication than colistin monotherapy (rifampicin, RR 1.31, 95% CI 1.01–1.69; fosfomycin, RR 1.23, 95% CI 1.01–1.53). There were no statistically significant differences between colistin monotherapy and any combination therapy (sulbactam, RR 0.74, 95% CI 0.41–1.34; carbapenem, RR 0.75, 95% CI 0.43–1.30; fosfomycin, RR 0.89, 95% CI 0.62–1.28).

## 4. Conclusions

This review provides a comprehensive and critical evaluation of the current evidence from meta-analyses on antibiotic combinations for the treatment of MDR Gram-negative bacteria. The relatively low number of patients included in these meta-analyses suggests that further appropriately designed studies should be conducted to evaluate the efficacy of combination therapy versus monotherapy. Several clinical studies have been conducted for the approval of β-lactamase inhibitor combination therapies, but fewer trials have targeted patients infected solely with MDR pathogens. Real-world data must be accumulated and analyzed to determine the efficacy of treatments for patients with MDR bacterial infections. Numerous in vitro and in vivo studies have evaluated the efficacy of antibiotics and antibiotic combinations in the treatment of MDR Gram-negative pathogens. However, clinical studies are limited, except those on colistin combination therapies. It is therefore necessary to collect further evidence on this topic. Combination therapies may prevent the emergence of resistance, achieve higher rates of success, and allow lower doses or shorter treatment periods, albeit with higher costs, potential side effects, and a potential for the emergence of a higher level of resistance than that predicted by in vitro studies.

## Figures and Tables

**Table 1 antibiotics-11-00524-t001:** The reports of systematic review and meta-analysis on antibiotic combinations.

Study	Year Published (Research Duration from Databases)	Database	Study Design Included in Meta-Analysis	Antibiotic Regimens (Monotherapy vs. Combination Therapy)	Type of Infection	Pathogens	Main Outcomes (Monotherapy vs. Combination Therapy)
Sfeir et al. [10]	2018 (up to 15 June 2017)	MEDLINE EMBASE Cochrane Library	pro or retrospective observational, cohort, and active surveillance	BL-BLI including piperacillin-tazobactam vs. carbapenem	BSI	ESBL-producing Enterobacterales	Mortality BL-BLI vs. carbapenem as definitive, OR 0.96, 95% CI 0.59–1.86, as empirical, OR 1.13, 95% CI 0.87–1.48 TAZ/PIPC vs. carbapenem as definitive, OR 0.97, 95% CI 0.59–1.6, as empirical, OR 1.27, 95% CI 0.96–1.66
Zhang et al. [11]	2021 (up to December 2020)	Cochrane Library PubMed EMBASE	RCT cohort	BL-BLI vs. carbapenem	cUTI APN	ESBL-producing Enterobacterales	mortality, RR = 0.63, 95% CI 0.30–1.32 clinical success, RR = 0.99, 95% CI 0.96–1.03 microbiological success, RR = 1.06, 95% CI 1.01–1.11
Sternbach et al. [12]	2018 (up to December 2017)	PubMed CENTRAL LILACS	RCT	CZA vs. comparator (mainly carbapenem)	cUTI cIAI NP	mostly Enterobacterales (~25% ESBL- carrying)	30-day mortality, RR 1.10, 95% CI 0.70–1.72 serious adverse events, RR 1.24, 95% CI 1.00–1.54
Che et al. [13]	2019 (up to December 2018)	Medline Embase Cochrane Library	RCT	CZA vs. carbapenem	cUTI APN	mostly Enterobacterales	clinical success, RD 0.00, 95% CI 0.06–0.06 microbiological success, RD 0.07, 95% CI 0.04–0.18 serious adverse events, RD 0.02, 95% CI 0.00–0.04
Isler et al. [25]	2020 (The dates of coverage were 27 January 2020 to 10 February 2020)	PubMed, CENTRAL, CINAHL, Scopus OvidMedline OvidEmbase Web of Science	RCT	CZA vs. carbapenem	cUTI cIAI HAP/VAP	ESBL and AmpC-producing Enterobacterales	clinical response for ESBL producers, RR 1.02, 95% CI, 0.97–1.08 for AmpC producers, RR, 0.91, 95% CI 0.76–1.10
Onorato et al. [15]	2019 (up to February 2019)	Medline Google Scholar Cochrane Library	cohort case-control case series	CZA vs. CZA plus other antibiotics	any	Carbapenem resistant Enterobacterales *P. aeruginosa*	mortality rate, RR 1.18, 95% CI 0.88–1.58 rate of microbiological cure, RR 1.04, 95% CI 0.85–1.28
Fiore et al. [16]	2020 (up to 2 February 2020)	Medline EMBASE CENTRAL	RCT cohort	CZA vs. CZA plus other antibiotics	any (mostly BSI)	Carbapenem resistant (mainly KPC producing) Enterobacterales	mortality rate, OR 0.96, 95% CI 0.65–1.41
Li et al. [17]	2021 (up to 31 March 2021)	PubMed EMBASE Web of Science CNKI Wanfang Data databases	cohort case series cross sectional	CZA vs. CZA plus other antibiotics	any	Any (carbapenem resistant)	overall mortality rates, OR 1.03, 95% CI 0.79–1.34 clinical success, OR 0.95, 95% CI 0.64–1.39 microbiologically negative, OR 0.99, 95% CI 0.54–1.81 posttreatment resistance of CZA, OR 0.65, 95% CI 0.34–1.26
Fiore et al. [18]	2021 (up to November 2020)	Medline EMBASE CENTRAL	retrospective cohort case-control	C/T vs. C/T plus other antibiotics	any	*P. aeruginosa* ESBL producing Enterobacterales	all-cause mortality, RR 0.31, 95% CI 0.10–0.97 clinical improvement, RR 0.97, 95%CI 0.54–1.74 microbiological cure, RR 0.83, 95%CI 0.12–5.70
Paul et al. [19]	2003 (up to March 2002)	Medline Embase Lilacs Cochrane Library	RCT	B-lactam vs. β-lactam-aminoglycoside combination	fever and neu-tropenia	any	all cause fatality, RR 0.85, 95% CI 0.72–1.02 treatment failure, RR 0.92, 95%CI 0.85–0.89 any adverse event, RR 0.85, 95%CI 0.73–1.00
Paul et al. [20]	2004 (up to March 2003)	Medline Embase Lilacs Cochrane Library	RCT	B-lactam vs. β-lactam-aminoglycoside combination	severe infections	any	all cause fatality, RR 0.90, 95% CI 0.77–1.06 clinical failure, RR 0.87, 95%CI 0.78–0.97 nephrotoxicity, RR 0.36, 95%CI 0.28–0.47
Zusman et al. [21]	2017 (up to 10 April 2016)	PubMed Cochrane Library	RCT retrospective observational	polymyxin monotherapy vs. polymyxin-based combination therapy	any	carbapenem-resistant or carbapenemase-producing Gram-negative bacteria	mortality, uOR 1.58, 95% CI 1.03–2.42 mortality compared with combination with TGC, AG and FOM, uOR 1.57, 95% CI 1.06–2.32 mortality for *K. pneumoniae* bacteremia, uOR 2.09, 95% CI 1.21–3.6
Samal et al. [22]	2021 (up to 31 December 2018)	PubMed Cochrane Library	RCT pro or retrospective observational	polymyxin monotherapy vs. polymyxin-based combination therapy	any	polymyxin-susceptible, carbapenem-resistant or carbapenemase-producing Gram-negative bacteria	mortality, RR 0.81, 95% CI 0.65–1.01 polymyxin-carbapenem combination in mortality, RR 0.64, 95% CI 0.40–1.03
Cheng IL et al. [23]	2018 (up to July 2018)	PubMed Embase Cochrane databases	RCT	colistin monotherapy vs. colistin-based combination therapy	any (mostly VAP)	carbapenem- resistant Gram- negative bacteria (mostly *A. baumanii)*	all-cause mortality, RR 1.03, 95% CI 0.89–1.20 infection-related mortality, RR 1.23, 95% CI 0.91–1.67 microbiologic response, RR 0.86, 95% CI 0.72–1.04
Vardakas KZ et al. [9]	2018 (up to November 2016)	PubMed Scopus	RCT	colistin monotherapy vs. colistin-based combination therapy	any (mostly VAP or BSI)	MDR or XDR Gram-negative bacteria (mainly *K. pneumoniae* or *A. baumanii*)	mortality, RR 0.91, 95% CI 0.81–1.02 mortality (in favor of combination with high-dose colistin), RR 0.80, 95% CI 0.69–0.93
Liu J. et al. [24]	2021 (up to March 2020)	PubMed Embase Cochrane Web of Science	RCT pro or retrospective observational	colistin monotherapy vs. colistin-based combination therapy	any (mostly VAP or BSI)	MDR or XDR *A. baumanii*	clinical improvement: RFP, RR 1.28, 95% CI 0.67–2.45; FOM, RR 1.08, 95% CI 0.76–1.53; sulbactam, RR 1.02, 95% CI 0.86–1.22 clinical cure: carbapenem, RR 1.34, 95% CI 0.92–1.95; sulbactam, RR 1.28, 95% CI 0.90–1.83 rate of microbiological eradication (in favor of combination with RFP or FOM), RFP, RR 1.31, 95% CI 1.01–1.69; FOM, RR 1.23, 95% CI 1.01–1.53

Abbreviations; BL-BLI, β-lactam and β-lactamase inhibitor combination; BSI, blood stream infection; ESBL, expended spectrum β-lactamase; OR, odds ratio; CI, confidence interval; RCT, randomized control trial; cUTI, complicated urinary tract infections; APN, acute pyelonephritis; CZA, ceftadizime avibactam; C/T, Ceftlozane-tazobactam; cIAI, complicated intra-abdominal infection; NP, nosocomial pneumonia; RR, risk ratio; RD, risk difference; uOR, TGC, tigecycline; AG, aminoglycoside; FOM, fosfomycin; unadjusted odds ratio;VAP, ventilator associated pneumonia; BSI, blood stream infection; MDR, multi drug resistant; XDR, extensively drug resistant; RFP, rifampicin.

**Table 2 antibiotics-11-00524-t002:** Meta-analyses evaluated the efficacy of carbapenem versus β-lactam and β-lactamase inhibitor combination therapies.

Study	No. of Studies Analyzed	Patients			Results with Significance
		Infections	Pathogens	Antibiotics	
Sfeir et al. [10]	25 (cohort or case-control)	BSI	ESBL-producing Enterobacterales	BL-BLI including PIPC/TAZ vs. carbapenem	None
Zhang et al. [11]	10 (3 RCTs and 7 cohort)	cUTI APN	ESBL-producing Enterobacterales	BL-BLI vs. carbapenem	None

Abbreviations; BL-BLI, β-lactam and β-lactamase inhibitor combination; PIPC/TAZ, piperacillin-tazobactam; BSI, blood stream infection; ESBL, expended spectrum β-lactamase; RCT, randomized control trial; cUTI, complicated urinary tract infections; APN, acute pyelonephritis.

**Table 3 antibiotics-11-00524-t003:** Meta-analyses evaluated the efficacy of carbapenem versus ceftazidime avibactam.

Study	No. of Studies Analyzed	Patients			Results with Significance
		Infections	Pathogens	Antibiotics	
Sternbach et al. [12]	7 RCTs	cUTI cIAI NP	mostly Enterobacterales (~25% ESBL-carrying)	CZA vs. comparator (mainly carbapenem)	Significantly higher rate treated with CZA (RR 1.24, 95% CI 1.00–1.54)
Che et al. [13]	3 RCTs	cUTI APN	mostly Enterobacterales	CZA vs. carbapenem	SAEs with CZA were numerically higher (RD = 0.02, 95% CI 0.00 to 0.04; *p* = 0.06).
Isler et al. [14]	5 RCTs	cUTI cIAI HAP/VAP	ESBL and AmpC- producing Enterobacterales	CZA vs. carbapenem	CZA showed a better microbiologic response for ceftazidime non-susceptible Enterobacterales (RR 1.21, 95% CI 1.07–1.37)

Abbreviations; CI, confidence interval; RCT, randomized control trial; cUTI, complicated urinary tract infections; APN, acute pyelonephritis; CZA, ceftazidime avibactam; cIAI, complicated intra-abdominal infection; NP, nosocomial pneumonia; ESBL, expended spectrum β-lactamase; SAE, severe adverse effect; RR, risk ratio.

**Table 4 antibiotics-11-00524-t004:** Meta-analyses evaluated the efficacy of ceftazidime avibactam (CZA) versus CZA combination therapy.

Study	No. of Studies Analyzed	Patients			Results with Significance
		Infections	Pathogens	Antibiotics	
Onorato et al. [15]	11 (cohort, case-control, case series)	any	CRE *P. aeruginosa*	CZA vs. CZA plus other antibiotics	None
Fiore et al. [16]	13 (7 RCTs, 6 cohorts)	any (mostly BSI)	CRE (mainly KPC producing)	CZA vs. CZA plus other antibiotics	None
Li et al. [17]	17 (11 cohort, 1 case series, 2 case-control, 3 cross sectional)	any	any (carbapenem resistant)	CZA vs. CZA plus other antibiotics	A trend of post-treatment resistance occurred more likely in CZA monotherapy (according to the pooled three studies, OR 0.18, 95% CI 0.04–0.78).

Abbreviations; CRE, carbapenem resistant Enterobacterales; RCT, randomized control trial; CZA, ceftazidime avibactam; OR, odds ratio; CI, confidence interval.

**Table 5 antibiotics-11-00524-t005:** Meta-analyses evaluated the efficacy of β-lactam versus β-lactam-AG combination.

Study	No. of Studies Analyzed	Patients			Results with Significance
		Infections	Pathogens	Antibiotics	
Paul et al. [19]	47 RCTs	fever and neutropenia	any	β-lactam vs. β-lactam-AG combination	Higher treatment success rate using combination therapy for the treatment of severe neutropenia (<100/mm3; RR 1.49, 95% CI 1.13–1.97) in both adults > 16 years old (RR 1.21, 95% CI 1.07–1.37) and children (RR 2.74, 95% CI 1.08–6.98).
Paul et al. [20]	64 RCTs	severe infections	any	β-lactam vs. β-lactam-AG combination	Clinical failure was more common with combination treatment overall (RR 0.87, 95% CI 0.78–0.97) Nephrotoxicity was significantly more common with combination therapy (RR 0.36, 95% CI 0.28–0.47).

Abbreviations; RCT, randomized control trial; AG, aminoglycoside; RR, risk ratio; CI, confidence interval.

**Table 6 antibiotics-11-00524-t006:** Meta-analyses evaluated the efficacy of polymyxin versus polymyxin combination therapy.

Study	No. of Studies Analyzed	Patients			Results with Significance
		Infections	Pathogens	Antibiotics	
Zusman et al. [21]	22 (RCT, retrospective observational)	any	CR or CP- GNB	polymyxin monotherapy vs. polymyxin-based combination therapy	Mortality rates were significantly higher with polymyxin monotherapy (OR 1.58, 95% CI 1.03–2.42)
Samal et al. [22]	39 (6 RCTs, 11 prospective and 22 retrospective observational)	any	polymyxin-susceptible, CR or CP GNB	polymyxin monotherapy vs. polymyxin-based combination therapy	Mortality rates were significantly lower with combination (OR 0.81, 95% CI 0.65–1.01)

Abbreviations; RCT, randomized control trial; CR, carbapenem resistant; GNB, Gram-negative bacteria; CP, carbapenemase producing; OR, odds ratio; CI, confidence interval.

**Table 7 antibiotics-11-00524-t007:** Meta-analyses evaluated the efficacy of colistin versus colistin combination therapy.

Study	No. of Studies Analyzed	Patients			Results with Significance
		Infections	Pathogens	Antibiotics	
Cheng IL et al. [23]	5 RCTs	any (mostly VAP)	CR-GNB (mostly *A. baumanii*)	colistin vs. colistin-based combination	None
Vardakas KZ et al. [9]	32 (3 RCTs, 6 prospective, 22 retrospective and one in both observational)	any (mostly VAP or BSI)	MDR or XDR-GNB (mainly *K. pneumoniae* or *A. baumanii*)	colistin vs. colistin-based combination	High-dose treatments (>6 million international units; RR 0.80, 95% CI 0.69–0.93), com-bination therapy was found to be significantly more effective in patients with bacteremia (RR 0.75, 95% CI 0.57–0.98)
Liu J et al. [24]	18 (7 RCTs, 11 retrospective)	any (mostly VAP or BSI)	MDR or XDR *A. baumanii*	colistin vs. colistin-based combination	Combination of RFP and FOM was associated with a significantly higher rate of microbiological eradication (RFP, RR 1.31, 95% CI 1.01–1.69; FOM, RR 1.23, 95% CI 1.01–1.53)

Abbreviations; RCT, randomized control trial; VAP, ventilator associated pneumonia; CR, carbapenem resistant; GNB, Gram-negative bacteria; RR, risk ratio; CI, confidence interval; BSI, blood stream infection; MDR, multi drug resistant; XDR, extensively drug resistant; OR, odds ratio FOM, fosfomycin; RFP, rifampicin.

## Data Availability

The data presented in this review are available upon reasonable request from the corresponding author.

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
