# Peer review of "Efficacy of Combination Therapies for the Treatment of Multi-Drug Resistant Gram-Negative Bacterial Infections Based on Meta-Analyses"

_antibiotics, 2022, doi:10.3390/antibiotics11040524_

Round 1
Reviewer 1 Report
This review is focused on summarizing the methods based on combination therapy to overcome the multidrug-resistant effect of gram-negative bacteria. This paper mainly discussed the antibiotic treatment targeting on β-lactamase production in combination with other antibiotics or non-antibiotics.
The review listed a substantial up to date studies which are sufficient. However, less of discussions, conclusions and perspective were provide by the authors.
Detail comments:
Why part 2 and part 3 have same title?
Lane 53: 2. Combinations of antibiotics plus drugs without antibiotic activity
Lane 240: 3. Combination of antibiotics plus drugs without antibiotic activity
Author Response
Manuscript ID: antibiotics-1665399
Title: Efficacy of combination therapies for the treatment of multi-drug resistant gram-negative bacterial infections based on meta-analyses
Journal: Antibiotics
Thank you for your reviewing our article. We made some changes in our manuscript according to reviewer’s suggestions with yellow highlight. We think some revises enhanced the quality of our manuscript.
Reviewer 1
Why part 2 and part 3 have same title?
Lane 53: 2. Combinations of antibiotics plus drugs without antibiotic activity
Lane 240: 3. Combination of antibiotics plus drugs without antibiotic activity
RESPONSE: Thank you for your comment. We changed the titles from “2. Combinations of antibiotics plus drugs without antibiotic activity” to “2. Combinations of antibiotics plus β-lactamase inhibitor” (Line 49) and from “3. Combination of antibiotics plus drugs without antibiotic activity” to “3. Antibiotics combinations” (Line 49, 248).
Reviewer 2 Report
Umemura et al. describes the significance and efficacy of combination therapy as a therapeutic strategy against gram-negative MDR pathogens based on previously reported meta-analyses.
A well organized and well-made review article that brings a good contribution to the current knowledge.
My only remark is - please clarify this:
"2. Combinations of antibiotics plus drugs without antibiotic activity"
"3. Combination of antibiotics plus drugs without antibiotic activity"
Author Response
Manuscript ID: antibiotics-1665399
Title: Efficacy of combination therapies for the treatment of multi-drug resistant gram-negative bacterial infections based on meta-analyses
Journal: Antibiotics
Thank you for your reviewing our article. We made some changes in our manuscript according to reviewer’s suggestions with yellow highlight. We think some revises enhanced the quality of our manuscript.
Reviewer 2
My only remark is - please clarify this:
"2. Combinations of antibiotics plus drugs without antibiotic activity"
"3. Combination of antibiotics plus drugs without antibiotic activity"
RESPONSE: Thank you for your comment. We changed the titles from “2. Combinations of antibiotics plus drugs without antibiotic activity” to “2. Combinations of antibiotics plus β-lactamase inhibitor” (Line 49) and from “3. Combination of antibiotics plus drugs without antibiotic activity” to “3. Antibiotics combinations” (Line 49, 248).
Reviewer 3 Report
In this review, "Efficacy of combination therapies for the treatment of multi-drug resistant gram-negative bacterial infections based on meta-analyses” Takumi Umemura et.al., presented a critical evaluation of the current evidence from meta-analyses on antibiotic combinations for the treatment of multidrug-resistant gram-negative bacteria.
The comments and suggestions for this manuscript are as follows-
- The introduction and the main body of the manuscript is a typical textbook type. This is lacking intellectual input from authors. The author should provide a comprehensive introduction.
- In addition to table-1, each subsection of meta-analysis needs a separate detailed table including the number of studies, groups, patients, and results with significance with proper references, for better understanding to the readers.
- In the manuscript, at several places, the author has included one group, two groups, or one trial for the meta-analysis. One example is page 6 line 121-127 “All-cause 30- day mortality was reported in six trials, which included one UTI, four intra-abdominal infection, and one pneumonia trial, that showed no significant difference in response between carbapenem and CZA (RR 1.23, 95% CI 0.87–1.76). Moreover, a microbiological response was reported in five trials, including three UTI, one intra-abdominal infection, and one pneumonia trial, that showed no significant difference between carbapenem and CZA (RR 1.04, 95% CI 0.93–1.17)”. How is the author concluding based on these limited studies?
Author Response
Manuscript ID: antibiotics-1665399
Title: Efficacy of combination therapies for the treatment of multi-drug resistant gram-negative bacterial infections based on meta-analyses
Journal: Antibiotics
Thank you for your reviewing our article. We made some changes in our manuscript according to reviewer’s suggestions with yellow highlight. We think some revises enhanced the quality of our manuscript.
Reviewer 3
- The introduction and the main body of the manuscript is a typical textbook type. This is lacking intellectual input from authors. The author should provide a comprehensive introduction.
RESPONSE: Thank you for your comment. We mentioned comprehensive features of the present review in the section of Introduction (Line 41-46).
- In addition to table-1, each subsection of meta-analysis needs a separate detailed table including the number of studies, groups, patients, and results with significance with proper references, for better understanding to the readers.
RESPONSE: Thank you for your advice. We added tables including detailed information of each meta-analysis in sections which we introduced two or more meta-analyses (Tables 2-7).
- In the manuscript, at several places, the author has included one group, two groups, or one trial for the meta-analysis. One example is page 6 line 121-127 “All-cause 30- day mortality was reported in six trials, which included one UTI, four intra-abdominal infection, and one pneumonia trial, that showed no significant difference in response between carbapenem and CZA (RR 1.23, 95% CI 0.87–1.76). Moreover, a microbiological response was reported in five trials, including three UTI, one intra-abdominal infection, and one pneumonia trial, that showed no significant difference between carbapenem and CZA (RR 1.04, 95% CI 0.93–1.17)”. How is the author concluding based on these limited studies?
RESPONSE: Thank you for your comment. In general, a meta-analysis focuses on certain population such as infection and bacteria. In CAZ section you pointed out, all meta-analyses focus on infections due to Enterobacterales. Therefore, we added the population which each meta-analysis focused on (Lines 79-80, 115, 165, 273, 281, 304, 330-331, 348-349, 375, 384-385 and 398). And the added tables according to your advice (your comment #2) also demonstrate the detailed populations (Table 2-7).
Round 2
Reviewer 3 Report
The author's response is satisfactory.